# Seeing is Solving: Unlocking Efficient Multimodal RL via View Alignment

**Qinsi Wang** [1][2]  **Jing Shi** [2]  **Kun Wan** [2]  **Handong Zhao** [2]  **Hancheng Ye** [1]  **Zishan Shao** [1]  **Jinghan Ke** [3]
**Yudong Liu** [1]  **Daniel Miranda** [2]  **Purvak Lapsiya** [2]  **Yiran Chen** [1]  **Wentian Zhao** [2]

## Abstract

Most existing methods simply port Large Language Model (LLM) RLFT techniques to VLMs, while ignoring a intrinsic property of multimodal models: their dynamic text–vision alignment. In this paper, we ask a new question: ***Can this intrinsic alignment be turned into a training signal that makes VLM RLFT more efficient?*** We analyze how a VLM plans to attend, actually attends, and ideally should attend during reasoning, and derive two lightweight metrics from these patterns. Predictive View Accuracy (PVA) estimates sample difficulty, and Reasoning View Accuracy (RVA) reflects the quality of chain-of-thought (CoT) reasoning. These alignment signals enable automated data curriculum and dense reasoning supervision. We introduce **FOCUS-RL**, a plug-and-play framework that can be seamlessly integrated into any VLM and dramatically boosts RLFT training efficiency. **FOCUS-RL** achieves $2.5\times - 4\times$ faster convergence over vanilla GRPO and consistent accuracy gains ($+4.4\%$ on average) across six different benchmarks.

## 1. Introduction

Vision-language models (VLMs) have achieved remarkable success across practical applications such as retrieval (Karpathy & Li, 2015; Faghri et al., 2017), visual question answering (Antol et al., 2015; Goyal et al., 2017), and content generation (Rombach et al., 2022; Saharia et al., 2022), exerting substantial influence on human life. Throughout the evolution of VLMs, ***text-vision alignment*** (Du et al., 2022; Long et al., 2022) has been widely recognized as a fundamental cornerstone for enhancing model performance (Li et al., 2021; 2022). From early cross-modal pretraining paradigms (Lu et al., 2019; Tan & Bansal, 2019) to CLIP's large-scale contrastive learning (Radford et al.,

2021; Jia et al., 2021), the core idea has consistently been to embed images and texts into a shared semantic space via a visual encoder, thereby unlocking powerful zero-shot generalization and transfer capabilities (Radford et al., 2021).

Building upon this foundation, researchers have further introduced ***Reinforcement Learning Fine-Tuning (RLFT)*** (Ouyang et al., 2022; Shao et al., 2024) to improve reasoning capabilities of VLMs, extending its exploration from language to multimodal settings (Yu et al., 2024; Zhang et al., 2025; Huang et al., 2025). For instance, VLM-R1 (Shen et al., 2025) leverages the GRPO paradigm to construct rule-verifiable rewards for multimodal reasoning, surpassing pure supervised fine-tuning (SFT) on various visual reasoning benchmarks. Similarly, RLHF-V (Yu et al., 2024; 2025c) utilizes human or AI feedback preferences in VLM reinforcement learning optimization, substantially reducing hallucinations during complex reasoning tasks and achieving significant gains on benchmarks like MMBench.

Although visual grounding annotations provide an explicit form of vision–text alignment that can guide models to learn more efficiently, obtaining such annotations at scale is prohibitively expensive. The diversity of visual question types and the sheer size of existing reasoning datasets make it impractical to manually annotate grounding for all training samples. Consequently, mainstream RLFT methods for VLM reasoning continue to inherit LLM-oriented training paradigms, relying primarily on token-level verifiable rewards such as answer correctness or preference judgments (Yu et al., 2024; 2025c; Huang et al., 2025), while largely ignoring visual alignment signals. This lack of vision-aware supervision forces VLMs to depend excessively on their language components, ultimately limiting their performance in complex visual or real-world scenarios (Li et al., 2023; Guan et al., 2024; Selvaraju et al., 2019).

To address this challenge, our work aims to answer the following central question: ***How can we effectively leverage text-vision alignment properties to make RLFT in VLM reasoning more efficient?*** We approach this systematically through three progressively deeper questions:

First, what types of text-vision alignment patterns arise in VLM reasoning? By analyzing attention distributions from

[1]Duke University [2]Adobe Inc. [3]UT Austin. Correspondence to: Wentian Zhao <wezhao@adobe.com>.

*Proceedings of the $43^{rd}$ International Conference on Machine Learning*, Seoul, South Korea. PMLR 306, 2026. Copyright 2026 by the author(s).

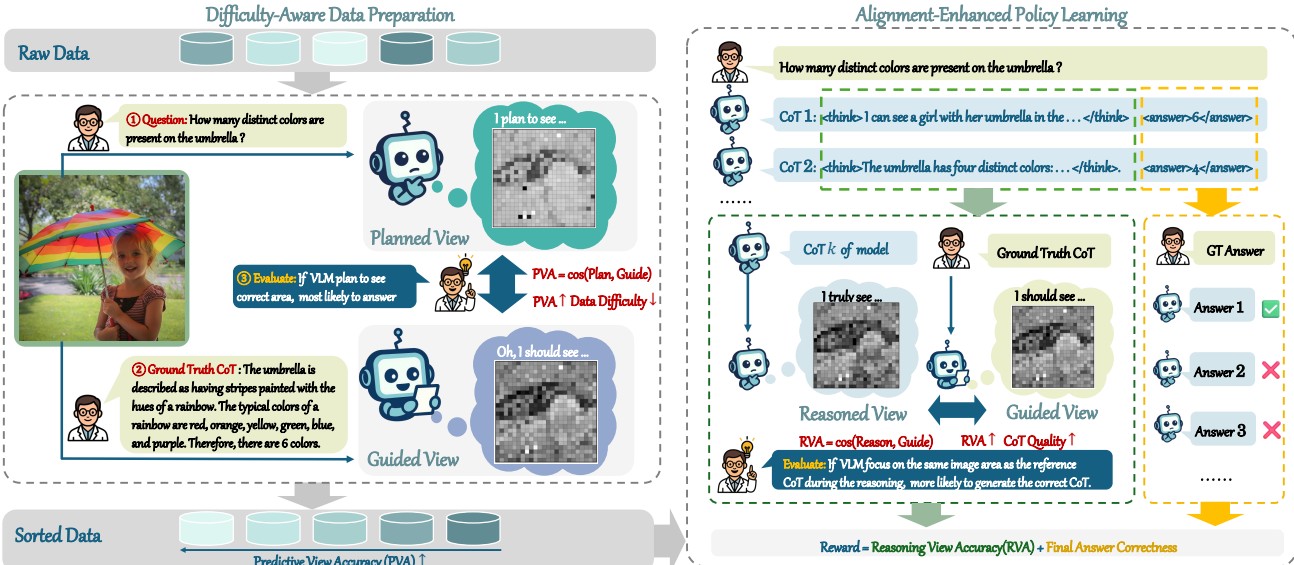

*Figure 1.* **Overview of FOCUS-RL Framework.** FOCUS-RL introduces two key enhancements over traditional RL frameworks. **(1) Data Processing Optimization (left)**: We employs the Predictive View Accuracy (PVA) metric to assess model-perceived sample difficulty before training, dynamically sorting data to enable the model's autonomous progression from simpler to more challenging samples. **(2) Policy Updating Optimization (right)**: We leverages the Reasoning View Accuracy (RVA) metric to evaluate the quality of the generated chain-of-thought (CoT), integrating both CoT quality and answer correctness into a denser reward signal. By combining these two optimizations, FOCUS-RL significantly improve training efficiency compared to vanilla RL frameworks.

text to images across various input conditions, we define three distinct patterns: *planned view*, *reasoned view*, and *guided view*. These three views patterns respectively capture where the model **plans** to look at, where it **actually** looks at, and where it **should** look at during the reasoning.

Secondly, how are these alignment patterns related to RLFT process? Through empirical analysis, we observe: **the more accurately a model looks at relevant image regions, the more likely it will answer correctly.** Motivated by this insight, we design two alignment-based metrics: *Predictive View Accuracy (PVA)* and *Reasoning View Accuracy (RVA)*, which quantify the accuracy of the model's planned view and reasoned view, respectively. Experimental results show that PVA can effectively predict the model's perceived difficulty of samples, while RVA can serve as a reliable evaluation metric for the quality of different CoT.

Finally, how can these insights accelerate VLM-RLFT? Leveraging above observations, we propose *FOCUS-RL, a plug-and-play efficient VLM RLFT framework that exploits text–vision alignment to enable automated curriculum learning and process supervision.* The framework of FOCUS-RL, illustrated in Fig. 1, involves optimization at two levels: At the data processing level, we employ PVA metric to predict the model's perceived difficulty of data samples. Based on these scores, training samples are pre-sorted prior to learning, enabling the model to autonomously learn progressively from simpler to more challenging samples at low cost. At the policy update level, we utilize RVA metric to evaluate the quality of different CoT. By jointly considering both CoT quality and final answer, we design a

denser reward that facilitates faster and more stable learning. Experimental results demonstrate that FOCUS-RL can significantly enhance model performance on reasoning tasks. Specifically, on Qwen2.5-VL-7B, FOCUS-RL achieves an average performance improvement of 4.4% over the original model and 2.6% over vanilla GRPO across seven reasoning benchmarks. Moreover, since FOCUS-RL relies solely on the alignment properties, it can be applied to arbitrary models and integrated as a plug-and-play component at only minor additional cost. Overall, our contributions include:

1. We introduce and define three text-vision alignment patterns: **planned view**, **reasoned view**, and **guided view**, capturing dynamic alignment patterns of what a VLM 'plans', 'actually' and 'should' to see.

2. We propose two alignment-based metrics, **Predictive View Accuracy (PVA)** and **Reasoning View Accuracy (RVA)**, experimentally validated to measure sample difficulty and CoT quality, respectively, thus establishing a direct connection between text-vision alignment and reinforcement learning processes.

3. We propose **FOCUS-RL**, a plug-and-play efficient VLM RLFT framework, which leverages PVA for autonomous curriculum learning and RVA for CoT process supervision, collectively achieving comprehensive acceleration. Experiments confirm that FOCUS-RL significantly enhances training efficiency over GRPOs.

## 2. Challenges and Background
RLFT for VLM reasoning has shown promising effectiveness; yet its substantial computational costs remains a major

obstacle to broader adoption. In this section, we examine two key sources of these costs and the associated challenges.

**Challenge 1: Low Sample Efficiency.** Current RLFT frameworks typically sample data randomly at each iteration and repeatedly train over the entire dataset (Shao et al., 2024; Yu et al., 2025b; Le et al., 2025). However, not every data point contributes meaningfully in every epoch (Le et al., 2025; Shi et al., 2025; Chen et al., 2025). For example, RL-ZVP (Le et al., 2025) shows that under GRPO setting, zero-gradient samples (i.e., samples where the model answers entirely correctly or incorrectly) constitute approximately 30%–99% of the training data throughout various stages, indicating a significant portion of samples provide no gradient signal to learning.

Autonomous curriculum learning has emerged as an effective strategy to address this inefficiency (Shi et al., 2025; Wang et al., 2025b). Unlike static curriculum learning that rely on a single ordering, autonomous curriculum learning dynamically sorts data online based on the model's inherent difficulty perception, allowing gradual progression from easy to hard samples. For instance, GainRL (Wang et al., 2025b) proposes angle concentration as an intrinsic signal derived, dynamically reordering and sampling training data in RLFT to improve sample and computational efficiency. AdaRFT (Shi et al., 2025) tracks recent model rewards to maintain a target difficulty threshold, selecting samples whose difficulty closely matches this threshold at each step.

However, these methods have so far only been applied to LLMs, and the proposed "difficulty-aware signals" have not been validated for VLMs. Therefore, a core challenge in adapting autonomous curriculum learning to VLM RLFT is: *How can we design VLM-specific signals to accurately estimate a model's perceived difficulty for a given example?*

**Challenge 2: Sparse Reward Signals.** In mainstream VLM RLFT settings, rewards typically rely on the verifiability of final answers (Huang et al., 2025; Shen et al., 2025; Yu et al., 2024), lacking metrics that measure the quality of intermediate steps. Therefore, reward signals tend to be sparse and coarse-grained, providing inadequate credit assignment to intermediate reasoning (Lightman et al., 2024; Zhang et al., 2025; Khalifa et al., 2025).

Process supervision mitigates this issue by evaluating intermediate CoT (Lightman et al., 2024; Luo et al., 2024; Tu et al., 2025). Numerous studies have demonstrated its effectiveness. For example, VisualPRM (Wang et al., 2025f) leverages large-scale multimodal preference ranking models (PRM) along with step-level annotated data to assign fine-grained scores to each reasoning step, significantly enhancing performance across multiple benchmarks. Similarly, URSA (Luo et al., 2025b) and PS-GRPO (Luo et al., 2025a) employ PRM-assisted online RL to explicitly penalize process inconsistency during training, outperforming vanilla

GRPO and achieving state-of-the-art results.

However, current process supervision methods predominantly focus on token-level step matching, necessitating extensive human or machine annotation of critical steps, thus hindering generalizability across different scenarios. A critical factor for enabling dense and generalizable reward signals in VLM RLFT is: *How can we leverage the inherent vision-specific characteristics of VLMs to estimate the quality of generated CoTs without step-level supervision?*

In the following sections, we will elaborate on how FOCUS-RL leverages the inherent text–vision alignment of VLMs to effectively address these two challenges.

## 3. Vision Alignment in RLFT

Due to the multimodal nature, text-vision alignment is critical to VLM performance. Prior studies have shown that the quality of image regions attended during reasoning strongly influences a VLM's accuracy, generalization, and robustness (Das et al., 2016; Selvaraju et al., 2019; Agrawal et al., 2018). For example, VQA-HAT (Das et al., 2016) demonstrates that misalignment between VQA model attention and human attention patterns negatively impacts model accuracy, whereas the HINT framework (Selvaraju et al., 2019) achieves significant performance gains by aligning model-generated importance maps with human-annotated regions. Additionally, benchmarks such as Flickr30k Entities (Plummer et al., 2015) and RefCOCO series (Yu et al., 2016; Mao et al., 2016) directly assess phrase grounding capabilities, while metrics like Pointing Games (Zhang et al., 2016) and Faithful & Plausible Visual Grounding (FPVG) (Reich et al., 2023) quantify whether models rely on genuinely relevant visual regions.

In this section, we comprehensively investigate how this alignment can inform and accelerate VLM training.

### 3.1. Dynamic View Patterns in VLM Reasoning

Notation. For training sample $d$ and a model $M$, suppose $d = (I_d, Q_d, \text{CoT}_d^{\text{gt}}, A_d^{\text{gt}})$ consists of an image $I_d$, a question $Q_d$, a reference chain-of-thought $\text{CoT}_d^{\text{gt}}$, and a ground truth answer $A_d^{\text{gt}}$. During the training, $M$ produces its own chain-of-thought $\text{CoT}_d^M$ and answer $A_d^M$ for sample $d$.

Definition of View. Assume model $M$ receives input content $S$, and $X = [x_1, \ldots, x_n] \in \mathbb{R}^{n \times h}$ denote the last-layer hidden states, where the first $m$ tokens $x_{1:m}$ correspond to the image and rest correspond to text. For any text token $x_i$ with $i > m$, its ***view*** over image tokens is defined as

$$v_i(S) = [\cos(x_i, x_1), \ldots, \cos(x_i, x_m)] \in \mathbb{R}^m,$$

where $\cos(x_i, x_j) = \langle x_i, x_j \rangle / (\|x_i\| \|x_j\|)$. Indeed, previous work has been demonstrated that $\cos(x_i, x_j)$ for $j > i$ accurately reflects the influence between tokens $i$ and $j$, and serves as a more precise indicator of token interactions than

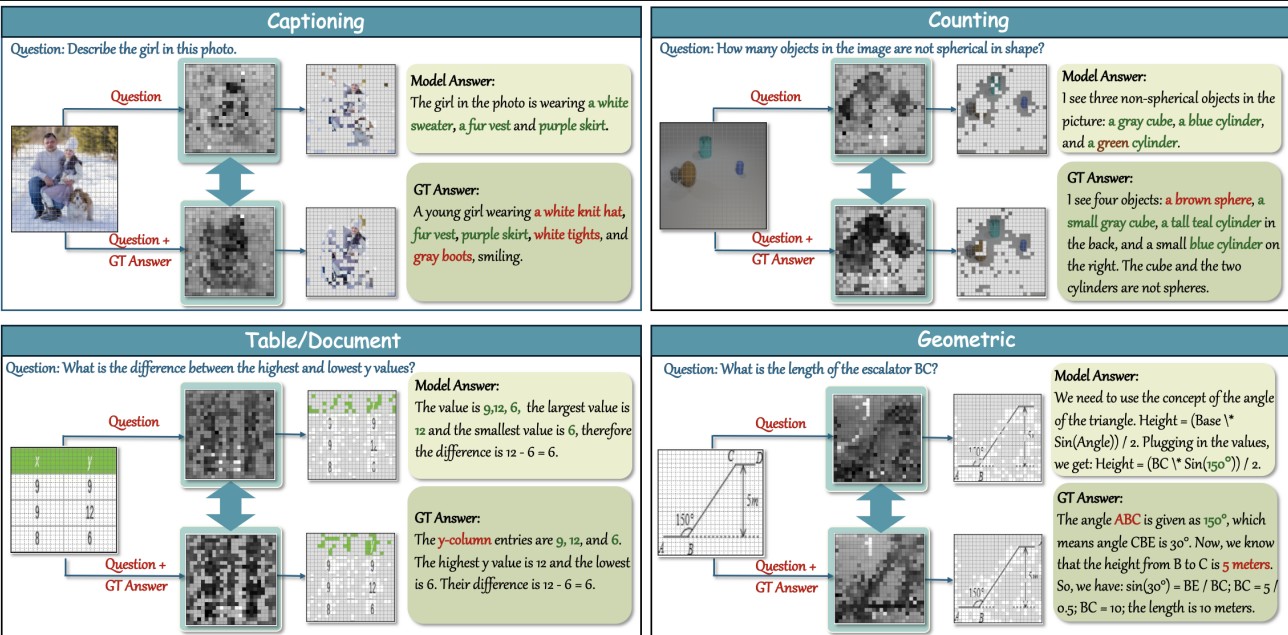

*Figure 2.* **Visualization of Planned Views and Guided Views across Different Reasoning Tasks.** To more intuitively illustrate the regions of the image the model attends to, we highlight the top 15% of image tokens for each view and overlay them on the original image. In the textual answers, we mark key terms correctly captured by the model in green, and key terms the model failed to capture but should have in red. Notably, the model typically answers based only on regions observed in its planned view, neglecting unobserved areas. Meanwhile, the guided view consistently demonstrates more accurate attention regions compared to the planned view.

traditional attention scores (Wang et al., 2024b; 2025c). Therefore, $v_i(S)$ captures how different image tokens influence model $M$ when generating token $i$, , effectively representing the alignment between token $i$ and image tokens $x_{1:m}$ and revealing image regions the model "views".

Based on the above definitions, we introduce three dynamic view patterns that emerge during VLM reasoning.

**Planned View.** The planned view refers to the model's anticipatory selection of image regions during the pre-filling stage. For model $M$ and input image-question pair $[I_d, Q_d]$, planned view is defined as view of final question token $n$,

$$v_d^{\text{plan}} := v_n([I_d, Q_d]) \in \mathbb{R}^m.$$

$v_d^{\text{plan}}$ predicts which image tokens the model $M$ will primarily utilize during decoding stage by examining the influence of image tokens on the first predicted token.

This early-stage prediction of relevant image tokens has been widely explored and validated in the domain on sparse inference. Numerous studies (Wang et al., 2025c; Yang et al., 2025; Endo et al., 2025) have demonstrated that by observing the influence of image tokens on the final question token, one can precisely capture the varying degrees of importance these image tokens hold toward the final answer. Furthermore, it has been shown that VLMs inference heavily relies on only a small subset of significant image tokens, retaining merely around 10% key tokens enables VLMs to perform nearly lossless inference (Wang et al., 2025c). We further empirically substantiate the predictive capability and significance of planned view in Sect. 3.2.

**Reasoned View.** The reasoned view corresponds to the image tokens actually utilized by the model during the reasoning process. For a given input $[I_d, Q_d]$, model $M$ generates $\text{CoT}_d^M$ and answer $A_d^M$. The reasoned view is defined as the average view over all tokens within $\text{CoT}_d^M$:

$$v_d^{\text{reason}} := \frac{1}{|\mathcal{T}_d^M|} \sum_{i \in \mathcal{T}_d^M} v_i([I_d, Q_d, \text{CoT}_d^M, A_d^M]) \in \mathbb{R}^m,$$

where $\mathcal{T}_d^M$ indexes the tokens of $\text{CoT}_d^M$ within all tokens. $v_d^{\text{reason}}$ measures the average impact of image tokens throughout the CoT generation process, revealing which parts of the image the model genuinely relies upon. In practice, due to token continuity, $v_d^{\text{reason}}$ is usually close to $v_d^{\text{plan}}$, while variations in the generated CoT responses to identical questions can introduce differences in $v_d^{\text{reason}}$.

**Guided View.** Now we know the planned and reasoned views of the model during reasoning, another critical and challenging question arises: *how can we determine the image regions the model should ideally focus on?* Annotating important image regions for each sample individually is both labor-intensive and time-consuming, and subtle differences exist in image regions attended to by different models.

To address this, we introduce the guided view, derived by directly providing the model with the ground truth CoT and observing its internal distribution. Formally, for a given model $M$, we input the image, question, reference CoT, and ground truth answer $[I_d, Q_d, \text{CoT}_d^{\text{gt}}, A_d^{\text{gt}}]$. The guided view

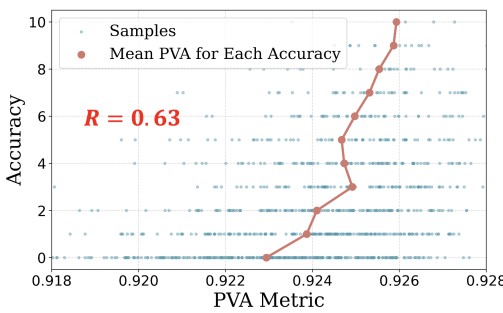

*Figure 3.* **Relationship between Model-Perceived Difficulty and PVA Score.** We randomly sample 1,000 examples and let the model to answer each example 10 times, recording the number of correct responses. The red curve denotes the average PVA score for samples at each correctness frequency. The Spearman correlation coefficient between PVA and answer correctness is 0.63.

is defined as the average view over all tokens within $\text{CoT}_d^{\text{gt}}$:

$$v_d^{\text{guide}} := \frac{1}{|\mathcal{T}_d^{\text{gt}}|} \sum_{i \in \mathcal{T}_d^{\text{gt}}} v_i\big([I_d, Q_d, \text{CoT}_d^{\text{gt}}, A_d^{\text{gt}}]\big) \in \mathbb{R}^m,$$

where $\mathcal{T}_d^{\text{gt}}$ indexes the tokens of $\text{CoT}_d^{\text{gt}}$ within all input tokens. In essence, the guided view simulates the model's attention distribution under the ideal scenario of generating a completely accurate reference CoT, revealing the regions the model should attend to through straightforward giving the reference answer. We further validate the effectiveness of the guided view in Sect. 3.2.

### 3.2. View-Driven Observations for Efficient RLFT

In this section, we distill these explorations into three observations that enhance view-driven training efficiency.

First, to verify whether the proposed view patterns accurately reflect the regions observed by the model, we conduct an intuitive test on the captioning task, where the model tends to describe exactly what it sees. As illustrated in the top-left corner of Fig. 2, the planned view focuses on the girl's clothing and skirt while overlooking her hat and boots. Correspondingly, the model's caption only mentions the clothing and skirt, omitting the other details. In contrast, the guided view exhibits a more complete focus on the girl, as the reference answer has a more accurate description.

We further apply the same visualization to other reasoning tasks including counting, table and geometric. Across these samples, we consistently observe that the model's answer is closely related to the salient image regions it attends to (highlighted in green), while elements it fails to attend to (in red) are often ignored, which in turn affects the reasoning process and final answer accuracy. Based on these findings, we can derive our first observation.

**Observation 1:** *The model tends to generate responses that focus on regions with high attention in its view, while ignoring low-attention regions.* As a result, if important visual evidence receives low attention, the response will incomplete or incorrect. We provide additional theoretical

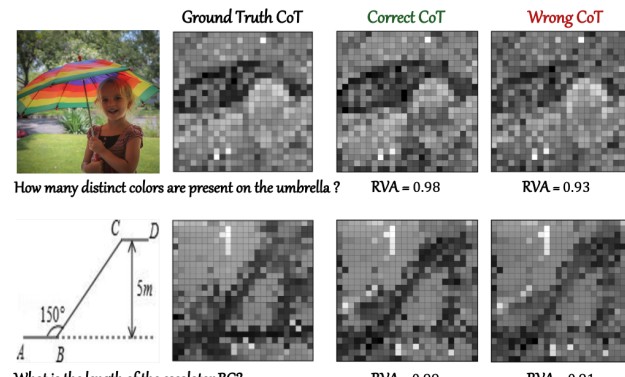

*Figure 4.* **Visualization of the Relationship Between Generated CoT Quality and RVA Score.** We visualize the model's reasoned views corresponding to CoTs of varying correctness, along with their associated RVA scores. Correct CoTs typically yield reasoned views that are more aligned with the guided view, leading to higher RVA scores. Additional examples are provided in Appendix B.2.

validation of the effectiveness of view in the Appendix A.1.

Further, to evaluate the accuracy of planned patterns, we define **Predictive View Accuracy (PVA)**, which measures similarity between the planned view and the guided view:

$$\text{PVA}(d) := \cos\big(v_d^{\text{plan}}, v_d^{\text{guide}}\big).$$

Higher PVA indicates greater alignment between planned and guided views, indicating model plan to see right regions.

To validate the relationship between PVA and sample difficulty, we randomly sample 1,000 examples from multimodal-open-r1-8k-verified dataset and prompt Qwen2.5-VL-3B to answer each question 10 times, recording the number of correct responses. Simultaneously, we compute the PVA for each sample. Experimental results are illustrated in Fig. 3, revealing a positive correlation between PVA and the accuracy of the model's responses. Furthermore, samples with higher response accuracy typically exhibit higher PVA values, suggesting that accurate visual planning is a necessary condition for correct reasoning. This finding suggests that PVA can serves as a reliable proxy for estimating the reasoning quality of a given model and samples. Since both planned and guided views can be computed with one inference, and the inputs used for calculating the guided view include those for planned view. PVA for a given sample $d$ can be computed in single inference.

**Observation 2:** *The similarity between planned and guided views (PVA) serves as an efficient predictor of model answer quality, and require only single inference.*

Finally, we examine the accuracy of the model's reasoned view during training. Similarly, we define **Reasoning View Accuracy (RVA)** to measure the similarity between the reasoned view and the guided view:

$$\text{RVA}(d) := \cos\big(v_d^{\text{reason}}, v_d^{\text{guide}}\big).$$

For classical RL training algorithms such as GRPO that

employ group normalization, the model $M$ generates multiple different CoT for a single sample $d$. To investigate the relationship between CoT quality and RVA, we randomly sampled examples from the multimodal-open-r1-8k-verified dataset and visualized the reasoned views of varying-quality samples, as shown in Fig .4. The results indicate that correct CoTs consistently focus on similar visual regions, thereby exhibiting higher RVA scores. Conversely, incorrect CoTs neglect essential visual details and have low RVA scores. This further confirms that accurately observing the correct regions is necessary for generating correct responses.

Importantly, RVA addresses the inherent difficulty in evaluating CoT by transforming the rigid constraint of *"matching exact CoT"* into a flexible yet rational constraint of *"matching critical visual regions observed by CoT"*, mitigates the risk of overfitting from overly strict constraints while preventing sparse rewards due to no process supervise.

**Observation 3:** *The similarity between the reasoned view and guided view (RVA) effectively reflects the quality of the model's generated CoT, elegantly transforming token-level CoT matching into view-level matching.*

## 4. FOCUS-RL Framework

In this section, we introduce **FOCUS-RL**, an efficient RL framework that leverages observations in Sect. 3 to accelerate RLFT. As illustrated in Fig. 1, the FOCUS-RL framework comprises two main components: Difficulty-Aware Data Processing and Alignment-Enhanced Policy Learning.

### 4.1. Difficulty-Aware Data Processing

To enhance training sample effectiveness, FOCUS-RL leverages the predictive capability of PVA, as identified in Observation 2, to achieve automated curriculum learning. Specifically, data processing consists of the following three parts:

**Data Sorting based on PVA.** Given a training dataset $\mathcal{D} = \{d_0, d_1, \ldots, d_N\}$, we compute the PVA value for each sample, $p_i := \text{PVA}(d_i) \in [-1, 1], \quad i = 0, \ldots, N$. Then, we sort the dataset based on the PVA values in descending order to obtain the sorted dataset $\mathcal{D}^{\downarrow\text{PVA}}$:

$$\pi = \underset{i \in \{0, \ldots, N\}}{\arg\text{sort}} (-p_i). \quad \mathcal{D}^{\downarrow\text{PVA}} := (d_{\pi(0)}, \ldots, d_{\pi(N)}).$$

Notably, as mentioned in Sect. 3.2, the computation of PVA score requires only a single inference. In contrast, previous curriculum learning methods typically assess sample difficulty using human annotators or full decoding processes with large models, which incurs substantial time and computational costs (Meng et al., 2025; Wang et al., 2025a; Deng et al., 2025a). For instance, calculating PVA values for 8,000 reasoning samples on a single NVIDIA A100 GPU takes less than one hour, whereas methods relying on human or large-model assessments, such as MM-Eureka (Meng et al., 2025) and MV-MATH (Wang et al., 2025a), requires several weeks to even months.

**Data Sampling based on Gaussian Probability.** During training, model progressively learns from simpler samples (higher PVA values) to more challenging ones (lower PVA values). At the $t^{th}$ iteration, we assign sampling probabilities to each data sample in $\mathcal{D}^{\downarrow\text{PVA}}$ based on a Gaussian distribution parameterized by $\mu_t$ and $\sigma_t$. A subset $d^{(t)}$ of size $n$ is sampled according to this probability distribution,

$$P_t(d_i^s) = \frac{1}{Z_t} \exp\left(-\frac{(i - \mu_t)^2}{2\sigma_t^2}\right), d^{(t)} \sim \text{Sample}(D_s; P_t, n),$$

where $Z_t$ is a normalization constant ensuring that probabilities sum to unity. Using probabilistic sampling instead of strict sequential sampling enhances training stability and robustness, and helps mitigate issues like data forgetting (Wang et al., 2025b).

**Probability Update based on Accuracy.** The Gaussian mean $\mu_t$ sets the peak sampling region. We initialize $\mu_0 = 0$ to prioritize data with high PVA values initially and then gradually increase $\mu_t$ as the model masters these samples, shifting focus toward more challenging, lower PVA-value data. At each iteration, we collect the average accuracy for all current training samples $d^{(t)}$, $\text{Acc}^{(t)} = \frac{1}{n} \sum_{i=1}^{n} \text{Acc}_{M_t}(d_i^{(t)})$, and update $\mu_t$ based on this value:

$$\mu_{t+1} = \mu_t + \frac{n}{2} \cdot \tanh\left(\alpha(\text{Acc}^{(t)} - \beta)\right),$$

where $\alpha$ adjusts accuracy sensitivity, and $\beta$ sets the target accuracy. This updating strategy preferentially selects samples close to the target accuracy, gradually incorporating more difficult, lower-PVA data, thereby enabling dynamic curriculum difficulty training.

Overall, to enhance training sample efficiency, **FOCUS-RL leverages PVA to predict the model's perceived difficulty of samples, which allows for cost-effective, automated curriculum training, effectively addressing Challenge 1.**

### 4.2. Alignment-Enhanced Policy Learning

To effectively supervise reasoning during training, FOCUS-RL leverages the RVA metric to evaluate the quality of the generated CoT, accelerating the training process. Policy learning in FOCUS-RL comprises two critical steps.

**CoT Evaluation via View Alignment.** Given a training sample $d$, the model $M$ generates $K$ candidate CoT-answer pairs under policy $\pi_\theta$: $(\text{CoT}_d^{(k)}, A_d^{(k)}) \sim \pi_\theta(\cdot \mid I_d, Q_d)$, where $(\text{CoT}_d^{(k)}, A_d^{(k)})$ denotes the $k$-th generated CoT and answer. To evaluate generated CoT, we record the corresponding reasoned view during model inference:

$$v_{d,k}^{\text{reason}} := \frac{1}{|\mathcal{T}_d^{(k)}|} \sum_{i \in \mathcal{T}_d^{(k)}} v_i([I_d, Q_d, \text{CoT}_d^{(k)}, A_d^{(k)}]),$$

where $\mathcal{T}_d^{(k)}$ represents the indices of tokens within $\text{CoT}_d^{(k)}$. Subsequently, we perform additional single forward pass with the ground-truth CoT to derive the guided view $v_d^{\text{guide}}$,

*Table 1.* **Performance Comparison with state-of-the-art RLFT Methods.** All models were evaluated using VLMEvalKit under same settings. FOCUS-RL(Data) considers only the Data Processing to ensure a fair comparison with curriculum learning baselines. FOCUS-RL(Data+Policy) incorporates both Data Processing and Policy Updating. FOCUS-RL outperforms all baselines on most tasks.

| Model | LogicVista | MathVerse | MathVista | MathVision | WeMath | DynaMath | Average |
|---|---|---|---|---|---|---|---|
| *Proprietary Vision-Language Models* | | | | | | | |
| GPT-4o | 45.9 | 50.2 | 61.4 | 30.4 | 40.0 | 32.3 | 54.6 |
| Gemini2.0-Flash | 56.2 | 54.4 | 73.4 | 41.3 | 57.1 | 43.7 | 61.4 |
| *Open Vision-Language Models Post-Training Method on Qwen2.5-VL-7B* | | | | | | | |
| Qwen2.5-VL-7B | 43.6 | 49.0 | 67.4 | 25.4 | 36.1 | 20.9 | 40.9 |
| OpenVLThinker-7B | 44.3 | 47.9 | 70.2 | 25.3 | 36.5 | 21.2 | 40.9 |
| MM-Eureka-7B | 46.9 | 50.3 | 73.0 | 26.9 | 36.2 | 24.2 | 42.9 |
| VLAA-Thinker-7B | 46.6 | 51.7 | 69.0 | 26.4 | 36.0 | 21.9 | 41.9 |
| MM-Open-R1(GRPO) | 44.2 | 52.1 | 71.6 | 27.3 | 38.5 | 22.3 | 42.7 |
| + AdaRFT | 45.0 | 52.8 | 71.8 | 27.3 | 37.8 | 23.9 | 43.1 |
| + GainRL | 45.7 | 53.0 | 72.2 | 28.1 | 39.2 | 24.3 | 43.8 |
| + FOCUS-RL(Data) | 47.4 | 53.8 | 73.0 | 28.6 | 40.0 | 25.3 | 44.7 |
| $\Delta_{\text{Qwen2.5-VL-7B}}$ | *+ 3.8* | *+ 4.8* | *+ 5.6* | *+ 3.2* | *+ 3.9* | *+ 4.4* | *+ 3.8* |
| $\Delta_{\text{Vanilla GRPO}}$ | *+ 3.2* | *+ 1.7* | *+ 1.4* | *+ 1.3* | *+ 1.5* | *+ 3.0* | *+ 2.0* |
| + FOCUS-RL(Data+Policy) | **49.1** | **54.2** | **73.5** | **29.0** | **40.0** | **25.8** | **45.3** |
| $\Delta_{\text{Qwen2.5-VL-7B}}$ | *+ 5.5* | *+ 5.2* | *+ 6.1* | *+ 3.6* | *+ 3.9* | *+ 4.9* | *+ 4.4* |
| $\Delta_{\text{Vanilla GRPO}}$ | *+ 4.9* | *+ 2.1* | *+ 1.9* | *+ 1.7* | *+ 1.5* | *+ 3.5* | *+ 2.6* |

and compute the RVA as the training reward of $\text{CoT}_d^{(k)}$:

$$r_{d,k}^{\text{CoT}} := \text{RVA}(d, k) = \cos\big(v_{d,k}^{\text{reason}}, v_d^{\text{guide}}\big).$$

Higher $r_{d,k}^{\text{CoT}}$ indicates stronger alignment between $\text{CoT}_d^{(k)}$ and $\text{CoT}_d^{\text{gt}}$, meaning the model's reasoning focuses on the correct regions of the image. Importantly, $v_{d,k}^{\text{reason}}$ is readily obtained during model inference, and the computation of the guided view requires only a single inference, resulting in negligible computational overhead for the overall training.

**Integrating Rewards for Policy Update.** During policy updates, we consider evaluation results from both CoT and answers. Assume the reward for answers is $r_{d,k}^{\text{ans}}$. We perform group normalization on both answer and CoT rewards to derive their respective advantages $\mathcal{A}_{d,k}^{\text{ans}}$ and $\mathcal{A}_{d,k}^{\text{CoT}}$. Then we compute the combined advantages to update policy:

$$A_{d,k} := A_{d,k}^{\text{ans}} + A_{d,k}^{\text{rva}}.$$

In summary, **FOCUS-RL addresses the reward sparsity inherent in RL (Challenge 2) by explicitly incorporating vision-based alignment measures between generated and reference CoTs,** effectively enhancing training efficiency.

## 5. Experiments

### 5.1. Experiment Setting

**Models and Datasets.** To comprehensively evaluate the effectiveness of FOCUS-RL, we conduct our primary experiments using the Qwen2.5-VL-7B model (Bai et al., 2025). Additionally, to verify the generalizability across different model sizes and architectures, we perform validations using three additional models: Qwen2.5-VL-3B (Team, 2025), InternVL3-3B (Wang et al., 2025d), and InternVL3-8B (OpenGVLab, 2025). For training, we utilize the multimodal-open-r1-8k-verified (LMMS-Lab, 2025) dataset, which contains approximately 8K multimodal rea-

soning samples accompanied by reference reasoning chains and verifiable answers. Detailed descriptions of datasets are provided in the Appendix C.1.

**Training and Hyperparameter Settings.** The hyperparameters in the FOCUS-RL training framework are derived from the accuracy sensitivity parameter $\alpha$ and the target accuracy $\beta$. Following prior work AdaRFT (Shi et al., 2025) and GainRL (Wang et al., 2025b), we set $\beta = 0.5$ to facilitate substantial loss and gradients during training, and $\alpha = 2$ to ensure the tanh function approximates linear behavior within the feasible ranges of Acc $\in [0, 1]$. Regarding model training configurations, we adopt GRPO as our baseline training framework. Each model undergoes 200 training iterations, with batch size of $n = 1024$. All experiments are conducted under the VLM-R1 framework (Shen et al., 2025) using eight NVIDIA A100 GPUs. Further training details are provided in the Appendix C.3.

**Baselines.** We compare our method against several state-of-the-art VLM RLFT approaches. In particular, OpenVLThinker-7B (Deng et al., 2025b; Deng & collaborators, 2025), MM-Eureka-Qwen-7B (Meng et al., 2025; ModalMinds, 2025), and VLAA-Thinker-7B (Wang et al., 2025e; UCSC-VLAA, 2025) are open-source models fine-tuned on Qwen2.5-VL-7B using high-quality human- or machine-annotated data. We directly evaluate these models as released. For comparison with curriculum-based RLFT methods, we reproduce AdaRFT (Shi et al., 2025) and GainRL (Wang et al., 2025b) under the same VLM-R1 training framework, using their proposed difficulty estimation strategies. All models are trained under identical settings to FOCUS-RL to ensure fair comparison.

*Table 2.* **Model Generalizability of FOCUS-RL.** We compare vanilla GRPO and FOCUS-RL using the same training and evaluation setting, detailed are provided in the Appendix C.3. $\Delta$ denotes the performance improvement of FOCUS-RL over vanilla GRPO.

| Model | LogicVista | MathVerse | MathVista | MathVision | WeMath | DynaMath | Average |
|---|---|---|---|---|---|---|---|
| *Qwen2.5-VL-3B* | 38.9 | 29.3 | 60.5 | 25.3 | 22.9 | 13.2 | 31.7 |
| + Vanilla GRPO | 39.1 | 31.9 | 62.2 | 27.1 | 24.2 | 14.8 | 33.2 |
| + FOCUS-RL(Data+Policy) | 41.0 | 34.1 | 64.5 | 28.8 | 27.1 | 16.1 | 35.3 |
| $\Delta_{\text{Vanilla GRPO}}$ | *+ 1.9* | *+ 2.2* | *+ 2.3* | *+ 1.7* | *+ 2.9* | *+ 1.3* | *+ 2.1* |
| *InternVL3-2B* | 33.8 | 23.5 | 57.3 | 21.7 | 22.4 | 14.6 | 28.9 |
| + Vanilla GRPO | 35.8 | 26.0 | 59.2 | 23.9 | 23.5 | 17.1 | 30.9 |
| + FOCUS-RL(Data+Policy) | 40.0 | 28.1 | 61.8 | 26.1 | 25.3 | 20.4 | 33.6 |
| $\Delta_{\text{Vanilla GRPO}}$ | *+ 4.2* | *+ 2.1* | *+ 2.6* | *+ 2.2* | *+ 1.8* | *+ 3.3* | *+ 2.7* |
| *InternVL3-8B* | 44.9 | 32.8 | 70.4 | 26.3 | 31.3 | 26.8 | 38.8 |
| + Vanilla GRPO | 45.4 | 35.4 | 72.1 | 29.2 | 32.9 | 29.3 | 40.7 |
| + FOCUS-RL(Data+Policy) | 47.1 | 37.1 | 73.0 | 31.9 | 34.1 | 31.2 | 42.4 |
| $\Delta_{\text{Vanilla GRPO}}$ | *+ 1.7* | *+ 1.7* | *+ 0.9* | *+ 2.7* | *+ 1.2* | *+ 1.9* | *+ 1.7* |

(a) Qwen2.5-VL-7B          (b) InternVL3-8B          (c) Completion Length

*Figure 5.* **Learning Dynamics of FOCUS-RL.** (a) and (b) illustrate the performance of checkpoints at different iterations on the MathVista for Qwen2.5-VL-7B and InternVL3-8B, respectively. (c) shows the average completion length of the model at different iterations during the training of Qwen2.5-VL-7B. FOCUS-RL demonstrates higher training efficiency and more stable outputs compared to GRPO.

## 5.2. Experiment Results

**Excellent Model Performance.** To assess and verify the efficiency of FOCUS-RL, we trained Qwen2.5-VL-7B models using different methods and evaluated their performance improvements across various evaluation sets. To ensure a fair comparison with baselines and explore the impact of different modules, we report model performance under two experimental conditions: (1) FOCUS-RL (Data), which only uses the difficulty-aware data processing module, and (2) FOCUS-RL (Data + Policy), which incorporates both data processing and policy updating strategies.

Experimental results, as presented in Tab. 1, demonstrate that FOCUS-RL (Data+Policy) achieves the best performance on most evaluation sets, achieving average accuracy improvements of 4.4% and 2.6% over the untrained and vanilla GRPO-trained models, respectively. Notably, on the LogicVista, FOCUS-RL outperforms the vanilla model by 4.9%. Moreover, FOCUS-RL (Data) also surpasses other curriculum learning baselines, such as AdaRFT and GainRL, whose evaluation metrics primarily target LLMs without considering VLM-specific characteristics. These results indicate that the autonomous curriculum learning and process supervision mechanisms provided by FOCUS-RL significantly improve VLM RLFT final performance, making it an effective plug-and-play acceleration component.

**High Hardware Efficiency.** To evaluate the hardware efficiency of FOCUS-RL, we tested the performance of checkpoints from different iterations trained on Qwen2.5-VL-7B and InternVL3-8B using both FOCUS-RL and vanilla GRPO on the MathVista dataset. As illustrated in Fig. 5 (a)(b), FOCUS-RL achieves the peak performance of vanilla GRPO within approximately 60 iterations for Qwen2.5-VL-7B and InternVL-8B, resulting in a 2.3× and 4× speedup, respectively. To further investigate training dynamics, Fig. 5 (c) presents the mean completion length of model-generated answers during training. Compared to vanilla GRPO, FOCUS-RL demonstrates more stable growth in answer length. This stability arises from FOCUS-RL autonomously transitioning from easier to harder samples, thereby enhancing learning capacity.

**Model Generalizability.** To validate the generalizability of FOCUS-RL, we post-trained Qwen2.5-VL-3B, InternVL3-2B, and InternVL3-8B on the multimodal-open-r1-8k-verified dataset and evaluated their performance across multiple test sets. Tab. 2 show that FOCUS-RL consistently improves performance over vanilla GRPO across all tested models. This generalizability is attributed to FOCUS-RL's reliance solely on text-vision alignment characteristics of VLMs, rather than model-specific features, allowing FOCUS-RL to be efficiently adapted to any model.

## 6. Conclusion

We introduce FOCUS-RL, a plug-and-play, efficient RLFT framework for VLMs that exploits their inherent text–vision alignment to enable autonomous curriculum learning and process supervision during training. Experiments show that FOCUS-RL substantially improves RLFT efficiency, delivering 2.5× and 4× speedups over GRPO across different VLMs. To our knowledge, this is the first study leverage text–vision alignment to accelerate RLFT for VLM.

## Impact Statement

This paper proposes FOCUS-RL, an efficient reinforcement learning fine-tuning (RLFT) framework for vision–language models (VLMs) that leverages text–vision alignment to enable autonomous curriculum learning and process supervision, substantially reducing training time and compute. The primary positive impact is to lower the cost and energy footprint of RLFT, improving accessibility for researchers and practitioners with limited resources and enabling faster iteration when adapting VLMs to downstream tasks.

At the same time, more efficient RLFT may also lower barriers to rapidly improving general-purpose VLM capabilities, which could increase the risk of misuse (e.g., scalable development of models that better support disallowed content generation, surveillance-related applications, or other harmful deployments). Our method does not introduce new data sources or explicitly expand model scope beyond standard RLFT, but it can accelerate capability amplification if applied without appropriate safeguards. We therefore encourage responsible use, including compliance with dataset licenses and privacy constraints, careful curation of preference/reward signals to avoid encoding harmful biases, evaluation for safety-related failure modes, and deployment with established monitoring and content-mitigation practices.

Overall, we view the expected benefits—reduced compute requirements and improved training efficiency for aligned VLM adaptation—as significant, while noting that the societal outcomes depend on how the resulting models are trained, evaluated, and deployed.

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

**Organization** In this Appendix, we provide in-depth descriptions of the materials that are not covered in the main paper, and report additional visualization and additional experimental results. The document is organized as follows:

## A. Theoretical Explanation

In our main experiments, we utilize the cosine similarity between the hidden states of text and image tokens as a proxy view to quantify the influence of image tokens on the generation of text tokens. In this section, we provide a theoretical justification for why cosine similarity serves as an effective and interpretable measure of inter-token interactions.

### A.1. Theoretical Explanation of View

In this section, we demonstrate that the cosine similarity between token representations directly correlates with the strength of the mapping between different outputs. Our analysis is extends upon theoretical of CoreMatching (Wang et al., 2025c).

**Notation.** Consider a standard pre-norm Transformer block at layer $\ell$ with residual stream (hidden state) $h_i^\ell \in R^d$ for token $i$. Let $Q^\ell, K^\ell, V^\ell, O^\ell$ be the learned projections for queries, keys, values, and the output map in the multi-head attention[1]. Define

$$q_i^\ell = Q^\ell h_i^\ell, \quad k_j^\ell = K^\ell h_j^\ell, \quad v_j^\ell = V^\ell h_j^\ell, \tag{1}$$

$$\alpha_{ij}^\ell = softmax_j \left( \frac{<q_i^\ell, k_j^\ell>}{\sqrt{d}} \right), \tag{2}$$

$$a_i^\ell = \sum_{j=1}^n \alpha_{ij}^\ell v_j^\ell, \quad z_i^\ell = O^\ell a_i^\ell. \tag{3}$$

We focus on the last text token $n$ (the one whose representation is read out to predict the next token/answer). Downstream, a linear readout $W_U$ converts $z_n^\ell$ to logits: for class/next-token $c$,

$$\ell_c = <w_c, z_n^\ell> = \sum_{j=1}^n alpha_{nj}^\ell <w_c, O^\ell v_j^\ell>, \tag{4}$$

where $w_c := W_U^\top e_c$., Eq. (4) makes explicit that the effective contribution of token $j$ to the output mapping is not $\alpha_{nj}^\ell$ alone; it is modulated by the directional alignment between $v_j^\ell$ and the readout direction $O^{\ell\top} w_c$.

---

[1]For clarity, we write a single-head derivation; the multi-head case follows by summing per-head contributions.

**Projection-guided influence.** Following the projection-guided criterion introduced for token importance, the influence of token $j$ on token $n$ at the representation level is measured by the projection of $v_j^\ell$ onto the attention score $a_n^\ell$:

$$Pi_{j \to n}^\ell := proj_{a_n^\ell}(v_j^\ell) = \|v_j^\ell\| \cos(\angle(v_j^\ell, a_n^\ell)). \tag{5}$$

This quantity captures the signed component of $j$ along the direction actually used by attention for $n$; tokens with large positive $\Pi_{j \to n}^\ell$ pull $a_n^\ell$ in their direction, thereby increasing their downstream effect.

**Two Assumption.** Empirical analyses in VLMs support the following layerwise regularities:

(A1) **Near-orthogonality of projections.** The maps $Q^\ell, K^\ell, V^\ell, O^\ell$ are approximately angle-preserving up to layer-wise scalars (i.e., their Gram matrices are close to scaled identities), and LayerNorm preserves direction.

(A2) **Diagonal-dominant self-attention for $n$.** The self term $\alpha_{nn}^\ell$ is typically (much) larger than most off-diagonal $\alpha_{nj}^\ell$, making the direction of $a_n^\ell$ dominated by $v_n^\ell$ with small corrections.

**Lemma 1 (Projection–Cosine equivalence).** Under Assumptions (A1)–(A2),

$$\Pi_{j \to n}^\ell \propto \cos(\angle(h_j^\ell, h_n^\ell)), \tag{6}$$

i.e., the projection-based influence in (5) is proportional to the cosine between the hidden states $h_j^\ell$ and $h_n^\ell$.

By (A2), $a_n^\ell = \sum_j \alpha_{nj}^\ell v_j^\ell \approx \alpha_{nn}^\ell v_n^\ell + \text{small}$, hence $\angle(v_j^\ell, a_n^\ell) \approx \angle(v_j^\ell, v_n^\ell)$ and $\Pi_{j \to n}^\ell \approx \|v_j^\ell\| \cos(\angle(v_j^\ell, v_n^\ell))$. By (A1) and LayerNorm, angles are preserved through $V^\ell$: $\cos(\angle(v_j^\ell, v_n^\ell)) \approx \cos(\angle(h_j^\ell, h_n^\ell))$. Absorbing slowly-varying norm factors into a global constant yields (6). □

**Theorem 1 (Cosine controls output mapping; attention alone does not).** Let $g_c^\ell := O^{\ell\top} w_c$ be the readout direction at layer $\ell$. Then token $j$'s contribution to the class-$c$ logit in (4) decomposes as

$$\Delta\ell_c(j \to n) = \alpha_{nj}^\ell \|v_j^\ell\| \|g_c^\ell\| \cos(\angle(v_j^\ell, g_c^\ell)). \tag{7}$$

Hence the sign and effective magnitude of $\Delta\ell_c(j \to n)$ are governed by a cosine term, not by the attention weight $\alpha_{nj}^\ell$ alone. Moreover, under Assumptions (A1)–(A2) and the standard observation that $g_c^\ell$ aligns closely with the direction of $a_n^\ell$ used to form $z_n^\ell$,

$$\cos(\angle(v_j^\ell, g_c^\ell)) \approx \cos(\angle(v_j^\ell, a_n^\ell)) \propto \cos(\angle(h_j^\ell, h_n^\ell)). \tag{8}$$

Therefore, ranking tokens by $\cos(\angle(h_j^\ell, h_n^\ell))$ is (up to a scalar) equivalent to ranking by the projection-guided influence $\Pi_{j \to n}^\ell$ that directly modulates logits via (4)–(7). □

**Implications for image–text interaction.** In a VLM where image and text tokens are co-attended, taking $j$ as an image token and $n$ as the last text token, the hidden-state cosine

$$\text{Score}_{\text{img} \to \text{text}}^\ell(j, n) := \cos(\angle(h_j^\ell, h_n^\ell)) \tag{9}$$

is a principled, architecture-aligned surrogate for the token's true downstream influence, because it tracks the projection $\Pi_{j \to n}^\ell$ that controls both the attention aggregate and the output logits. In contrast, the raw attention score $\alpha_{nj}^\ell$ does not encode alignment to the model's readout/aggregate direction and thus can overestimate tokens whose value vectors are nearly orthogonal to $a_n^\ell$ or $g_c^\ell$. Practically, computing $\cos(h_j^\ell, h_n^\ell)$ is inexpensive (no value/readout access required) yet better aligned with the model's computation of effective influence.

## B. Visualization

In the main text, we introduced three distinct types of view representations—planned view, reasoned view, and guided view—and analyzed three key observations that characterize the regularities and interaction patterns among these views. In this section, we further expand our empirical analysis by examining additional visual examples to validate and strengthen the observations presented earlier.

### B.1. Visualization of Planned and Guided View

As illustrated in Fig. 6, we showcase more examples highlighting both the planned view and guided view for various images. Across all cases, we observe that the planned view reliably captures the model's anticipated focus: it highlights the visual

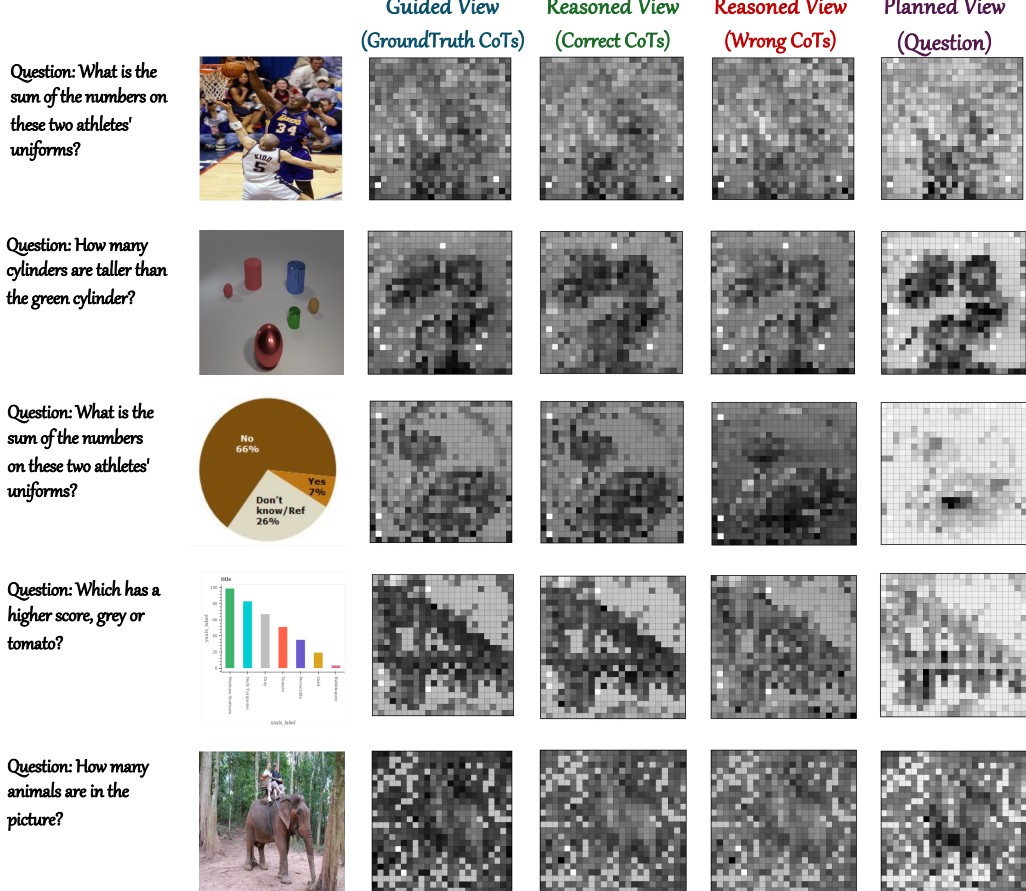

*Figure 6.* Visualization of planned view, guided view, and planned view for different image samples.

regions or objects the model intends to reference when generating an answer. In contrast, the guided view presents a broader and more comprehensive coverage of the image, encompassing all semantically important regions and cues that are relevant to the question. This observation aligns closely with Observation 1 and Observation 2 from the main paper—namely, that the similarity between the guided view and planned view serves as a strong indicator of whether the model is attending to the correct visual evidence, which in turn correlates with the likelihood of generating a correct answer.

### B.2. Visualization of Reasoned View

Moreover, we provide additional insights in Fig. 6 by visualizing two reasoned views corresponding to the same input sample: one where the model produces a correct chain-of-thought (CoT) and answer, and one where it fails. A clear pattern emerges—reasoned views that lead to correct answers tend to focus on visually coherent regions that are well-aligned with the guided view, indicating that the model is grounding its reasoning on the correct visual elements. On the other hand, reasoned views associated with incorrect answers often exhibit misaligned or diffuse attention, highlighting irrelevant or ambiguous regions. This misalignment hinders the model's ability to articulate an accurate reasoning path grounded in the correct visual content, ultimately resulting in an erroneous answer.

## C. Experiments Setting

In this section, we provide a comprehensive overview of the models, datasets, and baseline methods employed in our experiments, as well as the detailed training configurations used throughout our study.

## C.1. Models and Datasets

**Models.** We evaluate four open-source vision–language models: Qwen2.5-VL-3B, Qwen2.5-VL-7B (Bai et al., 2025), InternVL3-2B, and InternVL3-8B (Wang et al., 2025d). All four couple a high-resolution visual encoder with a general-purpose language backbone and accept interleaved image–text inputs, enabling end-to-end reasoning over charts, figures, and mathematical diagrams. The Qwen2.5-VL family targets a strong accuracy–efficiency trade-off: the 3B variant is optimized for lightweight inference on constrained hardware, while the 7B variant provides a higher-capacity baseline for multimodal reasoning and OCR-free perception. The InternVL3 family emphasizes fine-grained visual understanding and layout awareness; the 2B model provides a compact baseline, and the 8B model offers stronger reasoning capacity and improved robustness to diverse visual formats. Unless otherwise noted, we use the instruction-tuned checkpoints in their default modes.

**Datasets.** Our experiments cover six benchmarks designed to assess visual mathematical reasoning and logical inference: LogicVista (Xiao et al., 2024), MathVerse (Zhang et al., 2024), MathVista (Lu et al., 2023), MATHVision (Wang et al., 2024a), WeMath (Qiao et al., 2024), and DynaMath (Zou et al., 2025). Together they span symbolic and commonsense logic with visual cues (LogicVista), multimodal word problems that pair text with diagrams or plots (MathVerse), large-scale image-centric math questions requiring multi-step deduction (MathVista), extensions of math problem sets with accompanying figures or tables (MATHVision), curriculum-style problems across grade levels with diverse visual artifacts (WeMath), and dynamic or procedurally multi-stage tasks that stress compositional reasoning (DynaMath). For each dataset we follow the official splits and answer formats, focusing on concise, single-span solutions; this suite provides complementary coverage of algebraic, geometric, functional, and data-interpretation skills under realistic visual conditions.

## C.2. Baseline

We adopt three representative "thinking-style" LVLM baselines—OpenVLThinker-7B (Deng & collaborators, 2025), MM-Eureka-Qwen-7B (ModalMinds, 2025), and VLAA-Thinker-7B (Wang et al., 2025e)—as our multimodal reasoning baselines. OpenVLThinker-7B augments a Qwen2.5-VL-7B base with an iterative SFT→RL loop (GRPO) on distilled image–question–reasoning triplets, yielding chain-of-thought style visual reasoning and consistent gains on MathVista/MathVerse/MathVision. MM-Eureka-Qwen-7B trains on the curated MMK12 corpus and uses a two-stage recipe with rule-based reinforcement learning plus online filtering (and variants like ADORA/DAPO) to improve stability and accuracy on K-12 multimodal math. VLAA-Thinker-7B is built on the VLAA-Thinking dataset of verified step-by-step traces and an RL split, enabling a careful SFT vs. RL comparison and attaining strong results on OpenCompass-style multimodal reasoning benchmarks.

We also include two classic training-efficiency RFT baselines—AdaRFT (Shi et al., 2025) and GAIN-RL (Wang et al., 2025b)—as algorithmic baselines for reinforcement fine-tuning. AdaRFT applies adaptive-curriculum sampling driven by recent reward signals, plugging into PPO-style RFT to cut steps while boosting math-reasoning accuracy without changing model or reward design. GAIN-RL leverages a model-internal "angle concentration" signal (geometry of token hidden states vs. gradients) to prioritize learnable data and navigate RFT efficiently, improving efficiency with negligible overhead.

## C.3. Training Setting

To ensure effective and stable training of our vision-language model, we adopt the VLM-R1 framework as the foundational training baseline. Building upon this architecture, we train our proposed FOCUS-RL framework for a total of 200 iterations across multiple datasets and evaluate the resulting models to assess performance and generalization.

The training configuration of FOCUS-RL is carefully designed to align with standard practices in large-scale VLM training. Specifically, we employ a global batch size of 1024, calculated as `nproc_per_node (8) × gradient_accumulation_steps (16) × num_generations (8)`. This batch size is consistent with conventional settings in vision-language pretraining, which helps maintain training stability and ensures convergence. In addition to the batch size, all other training hyperparameters are kept consistent with the VLM-R1 configuration. Notably, FOCUS-RL is fully compatible with key optimization and acceleration techniques such as FlashAttention-2 and DeepSpeed ZeRO Stage 3, enabling efficient memory utilization and faster training on modern GPU infrastructures.

```
torchrun --nproc_per_node="8" \
    --nnodes="1" \
    --node_rank="0" \
```

```
  --master_addr="127.0.0.1" \
  --master_port="12349" \
src/open_r1/grpo_jsonl.py \
  --use_vllm False \
  --output_dir $OUTPUT_DIR/$RUN_NAME \
  --resume_from_checkpoint True \
  --model_name_or_path $model_path \
  --data_file_paths $data_paths \
  --image_folders $image_folders \
  --is_reward_customized_from_vlm_module $is_reward_customized_from_vlm_module\
  --task_type $TASK_TYPE \
  --per_device_train_batch_size 2 \
  --gradient_accumulation_steps 16 \
  --gradient_checkpointing true \
  --logging_steps 1 \
  --num_train_epochs 2 \
  --bf16 \
  --attn_implementation flash_attention_2\
  --run_name ${EXP_NAME} \
  --data_seed 42 \
  --save_steps 10 \
  --num_generations 8 \
  --max_completion_length 2048 \
  --reward_funcs accuracy format \
  --beta 0.04 \
  --report_to wandb \
  --dataset-name multimodal-open-r1-8k-verified \
  --deepspeed ${REPO_HOME}/zero3.json \
  --num_iterations 15
```

# D. Additional Experiments

## D.1. Cost Comparison

We estimate the computational costs of all methods following the protocol of Perception-R1 (Yu et al., 2025a). Specifically, we decompose the cost into data-preparation cost and training-time cost. For training time, we report GPU-hours, computed as the product of the number of GPUs and the effective training wall-clock time. To obtain comparable training costs for baselines that do not release full logs, we calibrate a per-sample cost from Perception-R1: their 1.4k RL samples require 167.4 A800-hours, which we treat as a unit cost and convert to A100-equivalent GPU-hours; assuming roughly linear scaling with the number of RL (and, when applicable, SFT) samples, we then estimate each baseline's training cost by multiplying this unit cost with its reported data volume. These estimates are intentionally coarse and are only used to compare orders of magnitude of compute; for methods with additional SFT stages or reward-model training, our numbers should be regarded as lower bounds, similar in spirit to the "w.h.p." bounds reported in Perception-R1.

*Table 3.* Computational costs comparisons between FOCUS-RL and representative baselines. w.h.p. stands 'with high probability'.

| Model | Training Time Cost (GPU-Hours) |
|---|---|
| Vision-R1 | $\geq$ 3400 A800-Hours (200K SFT + 10K RL) |
| MM-Eureka | $\approx$ 1800 A800-Hours w.h.p. (15K RL) |
| VLAA-Thinker | $\approx$ 3000 A800-Hours w.h.p. (25K RL) |
| OpenVLThinker | $\geq$ 3000 A800-Hours w.h.p. (25K SFT + RL) |
| R1-Onevision | $\geq$ 2900 A800-Hours w.h.p. (155K SFT + 10K RL) |
| R1-VL | $\geq$ 4100 A800-Hours w.h.p. (260K SFT + 10K RL) |
| MM-Open-R1 | $\approx$ 2880 A800-Hours w.h.p. (24K RL) |
| FOCUS-RL | $\approx$ 1152 A800-Hours w.h.p. (10K RL) |

The experimental results, as summarized in Tab.3, demonstrate the significant efficiency advantages of our proposed FOCUS-RL framework. Notably, FOCUS-RL achieves comparable or superior performance to existing state-of-the-art approaches, while requiring substantially fewer computational resources. Specifically, FOCUS-RL completes training using

approximately 1152 A100 GPU hours, in contrast to over 2000 A100 GPU hours required by other competitive baselines. This substantial reduction in training cost highlights the effectiveness of FOCUS-RL as a general and efficient solution for vision-language reasoning tasks. By maintaining high performance while drastically reducing resource consumption, FOCUS-RL offers a practical and scalable alternative for large-scale VLM training, particularly in scenarios where computational budgets are constrained. These results underscore the potential of FOCUS-RL to serve as a cost-efficient training paradigm for future multimodal reasoning systems.

### D.2. Signal Comparison

To further validate the effectiveness of our proposed framework in addressing the reward sparsity issue that commonly arises during training, we conduct a quantitative analysis of gradient activity throughout the learning process. Specifically, at various training epochs, we collect and examine the proportion of samples for which the model receives zero training gradient—that is, instances where the model's outputs are either entirely correct or entirely incorrect, resulting in no reward-based learning signal. As shown in Fig. 7, the original GRPO framework exhibits a significant proportion of such samples, particularly as training progresses.

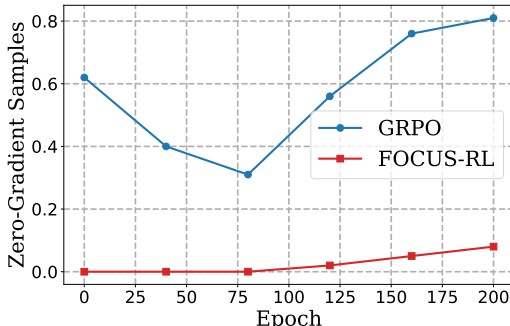

*Figure 7.* Comparison of reward sparsity during training using different methods.

This phenomenon is especially pronounced in later epochs, where the model increasingly produces fully correct answers for many samples. While this might superficially suggest performance improvement, it leads to a substantial underutilization of training resources, as these samples contribute no gradient for further learning and refinement. This inefficiency not only slows convergence but also limits the model's capacity to generalize effectively from partial understanding. In contrast, our FOCUS-RL framework mitigates this issue by leveraging a fine-grained evaluation of the quality of the model's chain-of-thought (CoT) reasoning. Instead of treating outputs as binary correct or incorrect, FOCUS-RL assigns dense and continuous reward signals based on intermediate reasoning quality. This ensures that even partially correct responses can provide meaningful gradient updates, enabling more efficient and informative training across all samples. The result is a more consistent learning signal and improved utilization of training data, which ultimately accelerates convergence and enhances model robustness.

