# OpenReview forum: "Seeing is Solving: Unlocking Efficient Multimodal RL via View Alignment"
_ICML.cc/2026/Conference — ICML 2026 regular_

### Official Review · Reviewer_RYkL · 2026-03-02

**Soundness:** 2
**Presentation:** 3
**Significance:** 2
**Originality:** 3
**Overall Recommendation:** 4
**Confidence:** 4

**Summary:**

This paper studies the visual reasoning limitations of multimodal large language models and argues that current models often fail to utilize visual information throughout the reasoning process. The authors propose a framework that enables iterative visual-grounded reasoning, allowing the model to repeatedly access visual features during multi-step reasoning. By maintaining visual grounding across reasoning steps, the method improves performance on complex visual reasoning tasks and demonstrates that sustained visual perception is critical for accurate multimodal reasoning.

**Compliance With Llm Reviewing Policy:**

Affirmed.

**Final Justification:**

Thanks for your rebuttal! I maintain my score.

**Key Questions For Authors:**

see the weaknesses

**Limitations:**

yes

**Strengths And Weaknesses:**

### Strengths
1. The motivation is clear. The two mentioned challenges are proper.
2. Good writing and presentation, the figure is nice, but the font size is kind of small.



### Weaknesses
1. The observation that visual info is getting lost during the reasoning chain is an uncountable new insight. And the idea of revisiting the visual info during the reasoning chain is also a famous topic.
2. The evaluated benchmarks mainly focus on math reasoning benchmarks; the authors can consider adding some more benchmarks in other fields.
3. There are some more methods that target sparse reward in visual reasoning (mentioned in Sec. 2), which should be discussed.

---

> ### Author Rebuttal · Authors · 2026-03-31
>
> We sincerely thank the reviewer for the time and effort spent on reviewing our paper. The concerns and questions raised are very helpful and have made our work more solid.
>
>
> ---
> ## 1. The observation that visual information is lost during long reasoning chains is not entirely new
>
> Thank you for pointing this out. We agree that visual information degradation during reasoning is a known phenomenon. Our core contribution is not discovering this issue, but rather **being the first to systematically formalize it into training signals for VLM RLFT**.
>
> Unlike existing methods that rely on architectural changes or explicit "look-back" modules, we introduce a fundamentally different view-alignment training framework:
>
> - **Novel Formulation:** We define three dynamic views (planned, reasoned, guided) and two lightweight metrics (PVA, RVA) that require zero human grounding annotations.
> - **Dual Training Signals:** These metrics directly translate visual alignment into (1) sample-difficulty signals for automatic curriculum learning, and (2) dense reasoning rewards for process supervision.
>
> FOCUS-RL clearly differs from existing visual-revisit methods in both problem formulation and technical design. We will explicitly highlight this distinction in the revised paper. Thank you again for this helpful suggestion.
>
> ---
> ## 2. The evaluated benchmarks mainly focus on math reasoning benchmarks
>
> We sincerely thank you for this suggestion. Due to the limited rebuttal time, we were not able to complete additional experiments on other tasks. However, **in the updated rebuttal and the revised version, we plan to include validation on a broader range of tasks, such as captioning and chart reasoning.** We will also more systematically discuss the applicability and limitations of FOCUS-RL in broader multimodal tasks, in noisier supervision settings, and in scenarios without high-quality reference CoT. We hope this will make the experimental conclusions more complete and more convincing.
>
> ---
> ## 3. There are more methods targeting sparse rewards in visual reasoning, which should be discussed
>
> Thank you very much for this valuable suggestion. Below, we provide a brief discussion of how FOCUS-RL differs from other methods that aim to address sparse rewards in visual reasoning.
>
> Existing work has started to alleviate the sparse-reward problem in visual reasoning, mainly along three directions. **The first direction is fine-grained preference supervision**. For example, RLHF-V [1] uses segment-level human correction for dense preference optimization, and V-DPO [2] uses visually guided preference learning to strengthen the model’s dependence on image content. **The second direction is process reward models (PRMs)**. For example, VisualPRM [3] trains a multimodal PRM to score intermediate reasoning steps. URSA [4] further integrate PRMs into online RL. **The third direction is explicit visual-anchor rewards**. For example, SATORI-R [5] turns captioning, localization, and final answers into verifiable rewards, while RewardMap [6] uses detailed rewards and multi-stage RL to reduce sparse rewards in fine-grained visual reasoning.
>
> Compared with these methods, FOCUS-RL is different in that it does not rely on an extra trained PRM, and it does not require additional bounding-box annotations or explicit intermediate labels. Instead, **it directly extracts training signals from the model’s own dynamic visual-text alignment**. In other words, while most existing methods improve supervision by adding an external reward model or extra intermediate labels, the core idea of FOCUS-RL is to convert the model’s internal visual alignment itself into RL supervision.
>
> We will add this analysis and discussion to the revised version. Thank you again for the helpful suggestion.
>
> [1] RLHF-V: Towards Trustworthy MLLMs via Behavior Alignment from Fine-grained Correctional Human Feedback.
>
> [2] V-DPO: Mitigating Hallucination in Large Vision Language Models via Vision-Guided Direct Preference Optimization.
>
> [3] VisualPRM: An Effective Process Reward Model for Multimodal Reasoning.
>
> [4] Unlocking Multimodal Mathematical Reasoning via Process Reward Model.
>
> [5] SATORI-R1: Incentivizing Multimodal Reasoning through Explicit Visual Anchoring.
>
> [6] RewardMap: Tackling Sparse Rewards in Fine-grained Visual Reasoning via Multi-Stage Reinforcement Learning.

---

> > ### Author Rebuttal · Reviewer_RYkL · 2026-04-03
> >
> > Thanks for your rebuttal! W `1` & `3` are addressed for my view. However, I expect some additional experiments for W `2` instead of only discussion.

---

> > > ### Author Response · Authors · 2026-04-05
> > >
> > > Thank you for the positive assessment of our work. We are very pleased that we were able to address the concerns raised in W1 and W3. Regarding W2, during the rebuttal period we further conducted experiments on the captioning task. The detailed experimental setup and results are as follows.
> > >
> > > **Experimental setup.** We performed a task-level ablation study of FOCUS-RL on the captioning task. Specifically, we conducted experiments on Qwen2.5-VL-3B, using COCO 2014 train as training set.
> > >
> > > For the **baseline (original GRPO)**, we designed the reward as $R_{\mathrm{ori}} = \mathrm{SPICE} + \mathrm{CIDEr}$ , where SPICE measures whether the generated caption is semantically faithful to the image content, and CIDEr evaluates how well the important phrases and linguistic patterns in the generated caption align with the reference captions.
> > >
> > > For **FOCUS-RL**, we computed the plan view, reasoned view, and guided view following the formulation in the paper. We first used PVA to estimate the difficulty of each sample for the model, and then ranked the samples by difficulty to enable adaptive curriculum learning. During inference, we further computed RVA, and the final training reward was defined as $R_{\mathrm{FOCUS\text{-}RL}} = R_{\mathrm{ori}} + 0.2\mathrm{RVA}$ . Notably, since captioning itself directly describes image content and the generated captions induce stable image-text attention distributions, **we do not require additional CoT for captioning task**. Instead, we can directly use the alignment between the reference caption and the image as the guided view, and the alignment between the model-generated caption and the image as the reasoned view.
> > >
> > > We trained the model with both original GRPO and FOCUS-RL for 100 iterations using a batch size of 1024, and then evaluated the resulting models on the test set.
> > >
> > > We evaluated our method on the CompreCap dataset (560 images, 412 object categories) using four metrics: Object Coverage (semantic alignment of generated noun phrases to ground-truth objects), Attribute and Relation Scores (0–5 ratings evaluating clause alignment with annotated attributes and subject-verb-object relations, respectively), and a Unified Score (a normalized, weighted aggregation of the prior three).
> > >
> > > | Model         | Object Coverage [0,100] | Attribute Score [0,5] | Relation Score [0,5] | Unified Score [0,100] |
> > > | ------------- | ----------------------- | --------------------- | -------------------- | --------------------- |
> > > | Qwen2.5-VL-3B | 72.3                    | 2.72                  | 2.76                 | 59.2                  |
> > > | +GPRO         | 74.1                    | 2.81                  | 2.78                 | 60.4                  |
> > > | +FOCUS-RL     | 77.2                    | 2.89                  | 2.88                 | 62.6                  |
> > >
> > > As shown in the table above, **FOCUS-RL also achieves clear performance gains on the captioning task, outperforming original GRPO**. This further demonstrates that FOCUS-RL can generalize beyond mathematical reasoning to other domains as well.
> > >
> > > We sincerely thank the reviewer for the time and effort devoted to evaluating our paper. The reviewer’s constructive feedback has helped make our work more complete, and we will incorporate the additional experiments conducted during the rebuttal period into the revised version of the paper. We hope that our response adequately addresses the reviewer’s concerns.

---

### Official Review · Reviewer_xQTc · 2026-03-11

**Soundness:** 3
**Presentation:** 3
**Significance:** 2
**Originality:** 2
**Overall Recommendation:** 3
**Confidence:** 4

**Summary:**

The paper studies reinforcement learning for VLMs and introduces text-vision alignment into the RL pipeline, improving the data selection and reward design of the traditional GRPO algorithm to encourage stronger attention to visual elements during VLM training. The authors propose two new metrics, PVA and RVA, which are used for training data ranking and as an additional reward signal, respectively. The motivation is intuitively reasonable, and the experimental results also demonstrate a certain level of effectiveness.

**Compliance With Llm Reviewing Policy:**

Affirmed.

**Key Questions For Authors:**

- Has the paper evaluated the sensitivity of the proposed method to the quality of the CoT? Also, in scenarios where CoT annotations are unavailable, would the method face significant limitations?

- There are some issues with writing errors. One obvious example is that the abstract states the method is evaluated on six different benchmarks, while the later introduction mentions across seven reasoning benchmarks; however, the table actually lists six benchmarks. This inconsistency may negatively affect the reader’s impression of the paper.

- The baselines used in the paper are relatively dated. It would be helpful to know whether the proposed method remains effective when compared against stronger and more recent baselines.

- The effectiveness of PVA and RVA still requires further clarification. In particular, the reported correlation between PVA and correctness is 0.63, which suggests some relationship, but is not especially strong.

**Limitations:**

yes

**Strengths And Weaknesses:**

This paper improves the effectiveness and efficiency of reinforcement learning for VLMs by introducing two text-vision alignment based metrics, PVA and RVA, for data selection and reward design, respectively.

**Strengths:**

- The paper addresses a clear and well-motivated problem: how to leverage the inherent visual alignment behavior of VLMs to improve RLFT efficiency. The paper is easy to follow and generally clearly written.

- Experimentally, the proposed method shows consistent performance gains over standard GRPO.

**Weaknesses:**

- The method relies on ground-truth CoT, which limits its applicability. The quality of the CoT may also affect the final performance, and collecting such CoT data can be cumbersome and costly.

- The evidence supporting the effectiveness of PVA and RVA is reasonable, but not yet fully convincing. For example, the reported correlation between PVA and correctness is 0.63, which indicates a meaningful relationship, but is still not particularly strong.

- The final accuracy improvement over vanilla GRPO is relatively modest.

---

> ### Author Rebuttal · Authors · 2026-03-31
>
> We sincerely thank the reviewer for the time and effort spent on reviewing our paper. The concerns and questions raised are very helpful and have made our work more solid.
>
> ---
> ## 1. Sensitivity of FOCUS-RL to the quality of CoT
>
> We thank the reviewer for raising this concern. We added additional experiments with reference CoTs of different quality to evaluate the robustness of FOCUS-RL to CoT quality.
>
> During the rebuttal, we further studied the sensitivity of FOCUS-RL to the quality of the reference CoT. Specifically, we sampled 5k training examples from multimodal-open-r1-8k-verified. Given the ground-truth answers, we used GPT-4o and Qwen2.5-VL-7B-Instruct to generate CoTs for these samples. In this way, we built three groups of reference CoTs: **the original reference CoTs from the dataset (high-quality), GPT-4o-generated CoTs (medium-quality), and Qwen2.5-VL-7B-Instruct-generated CoTs (low-quality)** . We then trained Qwen2.5-VL-3B under the FOCUS-RL framework for 50 epochs using each group of reference CoTs, and evaluated the trained models on the validation set.
>
> | Method             | LogicVista | MathVerse | MathVista | MathVision | WeMath | DynaMath |
> | ------------------ | ---------: | --------: | --------: | ---------: | -----: | -------: |
> | Qwen2.5-VL-3B      |       38.9 |      29.3 |      60.5 |       25.3 |   22.9 |     13.2 |
> | + Original GRPO    |       39.1 |      31.9 |      62.2 |       27.1 |   24.2 |     14.8 |
> | + Low-Quality CoT  |       39.5 |      33.7 |      63.9 |       28.1 |   26.3 |     15.0 |
> | + Medium-Quality CoT  |       40.2 |      34.4 |      64.5 |       28.6 |   27.1 |     15.8 |
> | + High-Quality CoT |       41.0 |      34.1 |      64.5 |       28.8 |   27.1 |     16.1 |
>
> As shown above, high- and medium-quality CoTs yield comparable gains. While low-quality CoTs slightly reduce improvements due to noisy visual guidance in $v_{\text{guide}}$, but they still clearly outperform standard GRPO. This confirms FOCUS-RL is highly robust to CoT quality.
>
> This robustness stems from our design: **(1) FOCUS-RL avoids rigid word-by-word imitation**, using $v_{\text{guide}}$ merely to align visual attention, meaning roughly correct CoTs suffice. **(2) $v_{\text{guide}}$ enables curriculum learning via difficulty estimation**, a unique advantage over text-only methods.
>
> Consequently,  for scenarios without CoT annotations, our experiments and the above analysis suggest that FOCUS-RL does not fail in such cases. Instead, CoTs can be generated automatically by a low-cost small model, and the model can still benefit from them. In addition, for tasks such as captioning, where the standard output is already long and contains rich visual information, the model output itself can often form a sufficiently accurate guided view even without explicit CoT. Therefore, FOCUS-RL can still work in such settings.
>
> ---
> ## 2. Issues with writing errors
>
> We thank the reviewer for the careful reading, and we apologize for the confusion caused by this inconsistency in the paper. The reviewer is correct. We will fix this issue in the revised version and carefully check the whole paper.
>
> ---
> ## 3. Baselines are relatively dated
>
> We thank the reviewer for this valuable suggestion. During the rebuttal, we added additional baselines based on text CoT supervision to further demonstrate the strong effectiveness of FOCUS-RL. **Please refer to our response to Reviewer AYCi, Q1**.
>
> ---
> ## 4. The effectiveness of PVA and RVA
>
> We thank the reviewer for the suggestion.
>
> Regarding the point that the correlation between PVA and correctness is 0.63 and may not seem very strong, we would like to clarify that PVA is not designed for perfect correctness prediction, but rather as an efficient difficulty estimator:
>
> - **Fast Prefilling-Stage Estimation:** PVA is computed entirely during prefilling, avoiding expensive decoding. Predicting final correctness before actual reasoning is inherently difficult, making a stable 0.63 correlation a remarkably strong signal of visual planning alignment. To our knowledge, no other prefilling metric achieves this.
> - **Sufficient for Curriculum Learning:** FOCUS-RL only requires a reliable *relative ranking* of sample difficulty, not absolute precision. Unlike static human or GPT-based coarse annotations, PVA provides a dynamic, model-specific estimate. For this curriculum ranking purpose, a 0.63 correlation is also highly effective.
>
> To further clarify the effectiveness of both PVA and RVA, during the rebuttal we also added more experiments and results (as mentioned above).
>
> We appreciate all the suggestions provided by the reviewer, and we will incorporate these experiments and analyses into the revised version.

---

> > ### Author Rebuttal · Reviewer_xQTc · 2026-04-03
> >
> > Thank the authors for the rebuttal; it has addressed most of my concerns. I will raise my scores.

---

> > > ### Author Response · Authors · 2026-04-03
> > >
> > > Dear Reviewer,
> > >
> > > We are very pleased that you feel we have addressed your concerns and you are willing to raise your score. We sincerely appreciate the time and effort you have devoted to reviewing our manuscript.
> > >
> > > We will incorporate all the additional experiments and discussions in the revised version. Your valuable suggestions have been extremely helpful in strengthening and solidifying this work.
> > >
> > > Sincerely,
> > >
> > > The Authors

---

### Official Review · Reviewer_AYCi · 2026-03-11

**Soundness:** 3
**Presentation:** 3
**Significance:** 3
**Originality:** 3
**Overall Recommendation:** 5
**Confidence:** 4

**Summary:**

This paper introduces PVA and RVA, two metrics that are used to improve multimodal RL training. The fundamental object underlying both of these metrics is the concept of a view, which is defined as a vector of cosine similarities between the hidden states of the final text token and the image tokens. This has been validated in prior work as an interpretable structure that measures the model’s visual attention. We assume we have access to a ground truth CoT and compute the view of the model after it, this is called the guided view. PVA attempts to capture the model’s perceived difficulty of a question and is defined by the similarity between the planned view, the model's view after getting the question, and the guided view. RVA is the similarity between the reasoned view, the model’s view after outputting its CoT, and the guided view, and it is used as a form of process supervision on the model’s response. PVA is used to rank samples for curriculum learning and RVA is used as process supervision during RL, which when combined boosts performance by 2.6% over the baseline.

**Compliance With Llm Reviewing Policy:**

Affirmed.

**Final Justification:**

The authors provided the key experiment I requested in the rebuttal so have raised my score to 5.

**Key Questions For Authors:**

“In practice, due to token continuity, v_reason is usually close to v_plan”. What does this mean? Do you show this empirically? If they are similar then why do we need both?


Figure 4, does “correct CoT” mean that these are CoT in cases where the answers are correct or that they are themselves are correct as reasoning traces, i.e. is the reasoning trace judged as being consistent with the answer?

**Limitations:**

As I mentioned, it should be acknowledged that requiring ground truth CoT is a limitation.

**Strengths And Weaknesses:**

**Strengths**

The topic area is very interesting. The paper is overall well written and easy to follow. The method is simple yet novel and I like that they ran experiments demonstrating it holds on other models. The figures are well-made and make it easy to understand the paper.

**Weaknesses**

A limitation is that ground truth CoT sequences are required. This is a limitation that is not really acknowledged as such in the paper and the authors should do so. The problem is that the purpose of regular RL is that we want the model’s reasoning traces to emerge organically, but we are putting a bias on them in this work. Furthermore, the reasoning traces that are used as supervision are themselves generated by a model, so they may be suboptimal.

Related to the above, I think there should be a text-only process supervision baseline. The paper claims that their method works because of multimodal alignment, but the effect of adding process supervision is not tested. If text-only supervision on the CoT traces works just as well, then there is no point to using their method. If this ablation is provided and FOCUS-RL maintains superior results, then I would raise my score.

---

> ### Author Rebuttal · Authors · 2026-03-31
>
> We sincerely thank the reviewer for the time and effort spent on reviewing our paper. The concerns and questions raised are very helpful and have made our work more solid.
>
> ---
> ## 1. Add text-only process supervision baselines
>
> We thank the reviewer for this suggestion. During the rebuttal, we added text-supervised CoT baselines to better demonstrate the advantage of FOCUS-RL. Specifically, we considered two classical baselines.
>
> **Baseline A: Auxiliary CoT imitation during RL.** Inspired by CHORD [1], we augment standard GRPO with auxiliary CoT supervision. Specifically, we compute rewards based solely on final answer correctness, while simultaneously applying a teacher-f orcing cross-entropy loss to the `<think>...</think>` tokens of reference paths. The final objective is:
> $L = L_{\text{GRPO}} + \lambda L_{\text{CoT-CE}}.$
> This baseline is used to test whether continuously adding CoT text supervision during training can improve the reasoning ability of VLMs. We set $\lambda = 0.1$.
>
> **Baseline B: Off-policy CoT inside the GRPO group (LUFFY-style).** Following LUFFY [2], we inject reference CoTs as expert off-policy trajectories directly into the GRPO group to test a more RL-native CoT integration. Each sample mixes standard on-policy rollouts with one reference CoT, evaluated by a shared reward (final answer correctness plus a minor format reward). We compute a mixed-group advantage to update on-policy trajectories via standard GRPO, alongside an advantage-weighted off-policy update for the reference CoT.
>
> We evaluate these classical text-only CoT baselines against FOCUS-RL under identical settings (100 epochs, Qwen2.5-VL-7B). As shown below, FOCUS-RL significantly outperforms both.
>
> | Method                              | LogicVista | MathVerse | MathVista | MathVision | WeMath | DynaMath |
> | ----------------------------------- | ---------: | --------: | --------: | ---------: | -----: | -------: |
> | Qwen2.5-VL-7B                       |       43.6 |      49.0 |      67.4 |       25.4 |   36.1 |     20.9 |
> | + Original GRPO                     |       44.2 |      52.1 |      71.6 |       27.3 |   38.5 |     22.3 |
> | + Baseline A |       45.8 |      51.5 |      73.4 |       26.8 |   39.4 |     23.0 |
> | + Baseline B |       45.3 |      52.6 |      71.9 |       27.7 |   40.0 |     24.2 |
> | + FOCUS-RL                          |       49.1 |      54.2 |      73.5 |       29.0 |   40.0 |     25.8 |
>
> FOCUS-RL’s superiority stems from two key advantages:
>
> - **Flexible Visual Grounding:** Rather than forcing strict text imitation, FOCUS-RL grounds reasoning in visual evidence while preserving textual flexibility.
> - **Curriculum Learning:** FOCUS-RL uses CoT to predict sample difficulty in advance, enabling efficient curriculum learning, a feature unattainable with pure text supervision.
>
> [1] On-Policy RL Meets Off-Policy Experts: Harmonizing Supervised Fine-Tuning and Reinforcement Learning via Dynamic Weighting. arXiv 2508.11408.
>
> [2] Learning to Reason under Off-Policy Guidance.  arXiv:2504.14945
>
> ---
>
> ## 2. Confusion about $v_{\text{reason}}$ and $v_{\text{plan}}$
>
> We apologize for the confusion. While $v_{\text{plan}}$ and $v_{\text{reason}}$ represent similar visual focus, they serve different stages due to their computational costs. **$v_{\text{plan}}$ is cheaply obtained via a single forward pass during prefilling**, making it ideal for efficient difficulty estimation (no decoding required). In contrast, **$v_{\text{reason}}$ averages attention across all generated CoT tokens.** Since full decoding is already required during RL training, computing $v_{\text{reason}}$ adds minimal overhead while accurately reflecting the actual reasoning process.
>
>  "$v_{\text{reason}}$ is usually close to $v_{\text{plan}}$" means decoding naturally follows the initial prefilling plan. We empirically verified this: on 100 random samples in multimodal-open-r1-8k-verified, the average cosine similarity between them is **0.965** on Qwen2.5-VL-7B. This is also supported by VLM pruning studies [3], confirming that key tokens identified during prefilling are sufficient for subsequent decoding.
>
> [3] LLaVA-PruMerge: Adaptive Token Reduction for Efficient Large Multimodal Models, arXiv 2403.15388
>
> ---
>
> ## 3. Confusion about “correct CoT” in Fig. 4
>
> We thank the reviewer for pointing this out. In Figure 4, "correct CoT" and "wrong CoT" simply refer to **model-generated** trajectories resulting in correct or incorrect final answers; they are not human-annotated or independently verified reasoning steps.
>
> Fig. 4 visualizes that correct answers typically align with relevant image regions (higher RVA), whereas incorrect answers deviate (lower RVA). This highlights why RVA provides effective fine-grained supervision in FOCUS-RL: instead of merely penalizing incorrect answers, it explicitly guides the model on *where it should look*. **We also analyzed the sensitivity of FOCUS-RL to CoT quality in response to Reviewer xQTc, Q1.**

---

> > ### Author Rebuttal · Reviewer_AYCi · 2026-04-02
> >
> > The authors provided the experiment I requested and their method maintains an advantage so I will raise my score as promised.

---

> > > ### Author Response · Authors · 2026-04-02
> > >
> > > Dear Reviewer,
> > >
> > > Thank you very much for your kind follow-up and for raising your score after reading our rebuttal. We are truly grateful that you feel we have addressed your concerns, and we sincerely appreciate your recognition of our work.
> > >
> > > We would also like to thank you again for the time and effort you invested in reviewing our paper, as well as for your thoughtful and valuable comments. Your feedback has been extremely helpful and has made this paper much stronger and more solid.
> > >
> > > We deeply appreciate your careful evaluation and support.
> > >
> > > Sincerely,
> > > The Authors

---

### Official Review · Reviewer_evKt · 2026-03-13

**Soundness:** 3
**Presentation:** 3
**Significance:** 3
**Originality:** 3
**Overall Recommendation:** 4
**Confidence:** 4

**Summary:**

This paper proposes FOCUS-RL, a reinforcement learning fine-tuning framework for vision-language models that explicitly exploits text-vision alignment during reasoning. The method defines three internal “views” derived from token-image interactions: planned view, reasoned view, and guided view. Based on these, the paper introduces PVA to estimate sample difficulty and drive curriculum learning, and RVA to score CoT quality and provide denser process-level rewards during RL. Experiments on several VLMs report both better final accuracy and substantially faster convergence.

**Compliance With Llm Reviewing Policy:**

Affirmed.

**Final Justification:**

Thanks for the rebuttal, which solves most of my concerns. I'd like to maintain the weak accept rating.

**Key Questions For Authors:**

My key questions focus on the effectivness of the proposed paradigm. Please refer to the weakness section.

**Limitations:**

Yes

**Strengths And Weaknesses:**

Strengths
------------
1. Clear multimodal motivation.
The paper identifies two genuine VLM-RLFT pain points and argues that existing solutions are largely inherited from LLM RL rather than tailored to multimodal reasoning. This motivation is well presented and technically relevant.

2. Conceptually neat formulation.
The planned/reasoned/guided view decomposition is intuitive, and PVA/RVA are simple, lightweight metrics that connect alignment behavior to curriculum learning and process supervision in a clean way.

3. Good empirical coverage.
The method is evaluated on multiple benchmarks and several base models, and the paper reports both final accuracy gains and convergence speedups.

Weaknesses
---------------
1. Limited evidence that view alignment causally improves reasoning.
The paper shows correlation between PVA and answer quality and visual examples linking RVA to better CoT, but the evidence is still largely correlational. The work would be stronger with stronger causal validation or more controlled ablations isolating whether the gains come from improved visual grounding versus simply adding another dense reward.

2. Evaluation could be narrow.
Most experiments are on a family of math/logic-oriented benchmarks. It remains unclear how well the method transfers to broader multimodal tasks, noisier supervision, or settings without high-quality reference CoT.

3. Dependence on reference CoT could be a practical limitation.
The policy-learning part requires ground-truth CoT to construct guided views. This may limit scalability in domains where answers are available but reliable reference reasoning traces are not.

---

> ### Author Rebuttal · Authors · 2026-03-31
>
> We sincerely thank the reviewer for the time and effort spent on reviewing our paper. The concerns and questions raised are very helpful and have made our work more solid.
>
> ---
>
> ## 1. Limited evidence that view alignment causally improves reasoning
>
> Thank you for this important comment.
>
> To verify that the benefit of FOCUS-RL does not come only from adding more supervision, but also from the visual alignment property of VLMs themselves, we added more baselines based on text CoT supervision during the rebuttal. The results are shown in the table above. In particular, **Auxiliary CoT imitation during RL** and **Off-policy CoT inside GRPO Group** are two classical text-CoT supervision methods. **Please refer to our response to Reviewer AYCi, Q1, for the detailed experimental settings.**
>
> | Method                              | LogicVista | MathVerse | MathVista | MathVision | WeMath | DynaMath |
> | ----------------------------------- | ---------: | --------: | --------: | ---------: | -----: | -------: |
> | Qwen2.5-VL-7B                       |       43.6 |      49.0 |      67.4 |       25.4 |   36.1 |     20.9 |
> | + Original GRPO                     |       44.2 |      52.1 |      71.6 |       27.3 |   38.5 |     22.3 |
> | + Auxiliary CoT imitation during RL |       45.8 |      51.5 |      73.4 |       26.8 |   39.4 |     23.0 |
> | + Off-policy CoT inside GRPO Group  |       45.3 |      52.6 |      71.9 |       27.7 |   40.0 |     24.2 |
> | + FOCUS-RL                          |       49.1 |      54.2 |      73.5 |       29.0 |   40.0 |     25.8 |
>
> The results show that FOCUS-RL clearly outperforms these pure text-CoT supervision methods. This suggests that the advantage of FOCUS-RL does not come only from extra dense supervision. More importantly, FOCUS-RL encourages the model to reason based on the correct visual evidence, while not forcing the model to follow a fixed textual reasoning form. In this way, it preserves the flexibility of textual reasoning while improving visual grounding.
>
> At the same time, we agree with your suggestion that stronger causal validation is needed. In the revised version, **we will add more strictly controlled causal experiments and ablation studies.** Thank you again for this valuable suggestion.
>
> ---
> ## 2. Dependence on reference CoT could be a practical limitation
>
> Thank you for raising this important concern. To directly address the issue of **reference CoT quality**, which you especially pointed out, we added additional experiments during the rebuttal.
>
> Given the ground-truth answers, we used GPT-4o and Qwen2.5-VL-7B-Instruct to generate CoTs for the samples. In this way, we built three groups of reference CoTs: the original reference CoTs from the dataset (high-quality), GPT-4o-generated CoTs (medium-quality), and Qwen2.5-VL-7B-Instruct-generated CoTs (low-quality). **Please refer to our response to Reviewer xQTc, Q1, for the detailed experimental settings.**
>
> | Method             | LogicVista | MathVerse | MathVista | MathVision | WeMath | DynaMath |
> | ------------------ | ---------: | --------: | --------: | ---------: | -----: | -------: |
> | Qwen2.5-VL-3B      |       38.9 |      29.3 |      60.5 |       25.3 |   22.9 |     13.2 |
> | + Original GRPO    |       39.1 |      31.9 |      62.2 |       27.1 |   24.2 |     14.8 |
> | + Low Quality CoT  |       39.5 |      33.7 |      63.9 |       28.1 |   26.3 |     15.0 |
> | + Mid Quality CoT  |       40.2 |      34.4 |      64.5 |       28.6 |   27.1 |     15.8 |
> | + High Quality CoT |       41.0 |      34.1 |      64.5 |       28.8 |   27.1 |     16.1 |
>
> These results demonstrate FOCUS-RL's robustness to CoT quality. While low-quality CoTs introduce noisy supervision that slightly destabilizes $v_{\text{guide}}$ (reducing overall gains compared to high/medium-quality CoTs), they still significantly outperform the GRPO baseline. **This confirms that FOCUS-RL does not rely on high-quality references and can achieve stable, clear improvements even with low-cost CoTs.**
>
> Based on these results, we further believe that, in settings without human CoT annotations, FOCUS-RL can still work by automatically generating CoTs with low-cost small models, and the target model can still benefit from them. In addition, for tasks such as captioning, where the standard output is already long and contains rich visual information, the output itself may be enough to form a reasonably accurate guided view even without explicit CoT.
>
> ---
> ## 3. Evaluation could be narrow
>
> We sincerely thank you for this suggestion. Due to the limited rebuttal time, we were not able to finish additional experiments on more tasks. However, **in the updated rebuttal and the revised version, we plan to add validation on a broader range of tasks, such as captioning and chart reasoning.** We will also more systematically discuss the applicability and limitations of FOCUS-RL in broader multimodal tasks in the revised verson.

---

> > ### Author Rebuttal · Reviewer_evKt · 2026-04-03
> >
> > Thanks for the rebuttal, which solves most of my concerns.

---

> > > ### Author Response · Authors · 2026-04-03
> > >
> > > Dear Reviewer,
> > >
> > > Thank you very much for your positive evaluation of our work. We are glad that you feel we have addressed most of your concerns.
> > >
> > > We sincerely appreciate the time and effort you have devoted to reviewing our manuscript. Your feedback has been very helpful in making this work more complete.
> > >
> > > Sincerely,
> > >
> > > The Authors

---

### Decision · Program_Chairs · 2026-04-30

**Decision:**

Accept (regular)

**Comment:**

This paper introduces FOCUS-RL, a framework that leverages internal text-vision alignment via PVA and RVA metrics to significantly boost VLM-RLFT efficiency. While reviewers initially highlighted concerns regarding the dependence on reference CoT data, the lack of causal evidence for visual grounding, and limited task diversity, the authors provided a robust rebuttal. By adding text-only supervision baselines, testing sensitivity to CoT quality, and providing new results on image captioning, the authors successfully demonstrated that FOCUS-RL provides unique multimodal advantages and broad generalizability. Given that the rebuttal fully satisfied all reviewers and confirmed significant empirical gains including 2.5x-4x faster convergence, this paper is recommended for acceptance.